# Studies on the Kinetics of Doxazosin Degradation in Simulated Environmental Conditions and Selected Advanced Oxidation Processes

**Joanna Karpinska \***[ID]**, Aneta Sokol, Jolanta Koldys and Artur Ratkiewicz**[ID]

Department of Biology and Chemistry, University of Bialystok, Ciołkowskiego 1K, 15-245 Białystok, Poland; anetka_w@uwb.edu.pl (A.S.); jolantka-93@wp.pl (J.K.); artrat@uwb.edu.pl (A.R.)

\* Correspondence: joasia@uwb.edu.pl

**Abstract:** The photochemical behavior of doxazosin (DOX) in simulated environmental conditions using natural waters taken from local rivers as a solvent was studied. The chemical characteristics of applied waters was done and a correlation analysis was used to explain the impact of individual parameters of matrix on the rate of the DOX degradation. It was stated that DOX is a photoliable compound in an aqueous environment. Its degradation is promoted by basic medium, presence of environmentally important ions such as $Cl^-$, $NO_3^-$, $SO_4^{2-}$ and organic matter. The kinetics of DOX reactions with $OH^-$ and $SO_4^-$ radicals were examined individually. The $UV/H_2O_2$, classical Fenton and photo-Fenton processes, were applied for the generation of hydroxyl radicals while the $UV/VIS:Fe_2(SO_4)_3:Na_2SO_2$ system was employed for production of $SO_4^-$ radicals. The obtained results pointed that photo-Fenton, as well as $UV/VIS:Fe_2(SO_4)_3:Na_2SO_2$ are very reactive in ratio to DOX, leading to its complete degradation in a short time. A quantitative density functional theory (DFT) mechanistic study was carried out in order to explain the molecular mechanism of DOX degradation using the GAUSSIAN 09 program.

**Keywords:** doxazosin maleate; advanced oxidation processes; hydroxyl radical; sulfate radical; photodegradation; DFT study

## 1. Introduction

Recent environmental studies show an appearance of new atypical compounds in aquatic ecosystems on a global scale. Called Emerging Organic Contaminants (EOC), they are created by hundreds of organic compounds belonging to different chemical classes [1]. Some of them are natural components of an environment, making their presence detectable due to advancements in sample preparation procedures [2–4], as well as new detection techniques [5–7]. They have been detected in clean surface waters at few ng $dm^{-3}$ levels while in polluted waters in the range from a few to hundreds of μg $dm^{-3}$ [8,9]. Many EOC-s compounds do not cause acute toxicity, but their presence in the environment entails a number of adverse changes, including interference in animal as well as human endocrine systems [1]. Compounds that exhibit such activity or are suspected of it are named Endocrine Disrupting Compounds (EDC). According to the definition given by The Endocrine Society, EDCs are: "an exogenous chemical, or mixture of chemicals, that interferes with any aspect of hormone action" [10]. One of the more numerous groups belonging to EDCs are traces of pharmaceuticals [1,8]. The main identified source of pharmaceuticals in surface freshwater environments are from wastewater treatment plants (WTP) [11]. Some compounds from the EDC-s group, especially pharmaceuticals, possess biocidal properties or are resistant to biodegradation, so the WTP-s based on activated sludge technology are unable to remove all of them [12]. Therefore, the search for improvements that can be

made to the water purification process is still currently a problem. The following modifications of the water technology were developed and introduced into practice: membrane bioreactors, purification ponds with aquatic plants, application of new microorganisms, enzymatic treatments [12] or advanced oxidation processes (AOPs) [13,14]. The advanced oxidation units are based on oxidation reactions of reactive chemical species such as hydroxyl, sulfate, chlorine and other radicals with organic pollutants [13,14]. Although AOPs are considered to be the most effective way for water treatment, their efficiency depends on many factors such as type of process, type and composition of the polluted water, and chemical properties of degraded contaminants [15,16]. Additionally, cost of operation as well as environmental implications should be considered [16]. Therefore, the introduction of AOP into the treatment process requires the optimization of the chemical conditions of an applied reaction and recognition of the chemical behavior of a main organic pollutant.

This paper presents the results of studies on kinetics of doxazosin (DOX) degradation under influence of light and some selected AOPs. Doxazosin mesylate [(4-amino-6,7-dimethoxy-2-quinazolinyl)-4-(1,4-benzodioxan-2-yl-carbonyl)-piperazine monomethansulphonate] (Figure 1) belongs to the group of $(\alpha_1)$-adrenoreceptor antagonists [17]. It is used for the treatment of benign prostatic hyperplasia [17,18] and blood hypertension [18]. It is well absorbed by the digestive tract after oral administration. Afterwards, it is partially metabolized and excreted with urine in the unchanged form (about 4.8%) and in the form of metabolites: products of demethylation (23%) and hydroxylation (12%) [19]. To the best of our knowledge, DOX chemical behavior under the influence of light or AOPs has not been reported yet. Among many available AOPs reactions, the runs of UV/Vis direct photolysis, UV-$H_2O_2$, classical and photo-Fenton processes, and oxidation by $SO_4^{\cdot}$ were studied [16]. Attempts have been made to assess the persistence of DOX in aquatic environment and indicate the environmental factors affecting the rate of its vanishing. For this purpose, DOX degradation rate under irradiation of sunlight in the presence of natural matrix was determined.

**Figure 1.** Chemical structure of doxazosin mesylate.

## 2. Materials and Methods

### 2.1. Chemicals

Doxazosin mesylate, DOX (Sigma-Aldrich, Germany), a stock solution at the concentration $2 \times 10^{-3}$ mol dm$^{-3}$ was prepared by dissolving an appropriate weight in 25 mL of MilliQ water. Working solutions at the concentrations $1.0 \times 10^{-6}$, $2.5 \times 10^{-6}$, $5.0 \times 10^{-6}$, $10^{-5}$, $1.5 \times 10^{-5}$ and $2 \times 10^{-5}$ mol dm$^{-3}$ were prepared by dilution in MilliQ water. Ferric sulfate (Fe$_2$(SO$_4$)$_3$) and anhydrous sodium sulfite (Na$_2$SO$_3$) were purchased from Chempur, Poland. Standard solution of ferric sulfate ($2.5 \times 10^{-3}$ mol dm$^{-3}$) was freshly prepared every day by the dissolution of an exact weighted amount in 50 mL of MilliQ water. Stock solution ($5 \times 10^{-3}$ mol dm$^{-3}$) of anhydrous sodium sulfite was freshly prepared every day from the pure product by dissolving an appropriate amount in 100 mL of MilliQ water.

Acetonitrile and methanol of HPLC grade were supplied by Sigma-Aldrich.

Tert–butyl alcohol (TBA) were purchased from Honeywell, Riedel-de Haën$^{TM}$.

Hydrogen peroxide (CHEMPUR, Poland) at the concentration of $10^{-1}$, $5 \times 10^{-2}$, $10^{-2}$, $2 \times 10^{-2}$, $2 \times 10^{-3}$ and $10^{-3}$ mol dm$^{-3}$ were prepared daily by suitably diluting its 30% solution in MilliQ water.

Other reagents used were: Concentrated acetic acid (Sigma-Aldrich, St. Louis, MO, USA), concentrated ammonium (Sigma-Aldrich), sodium hydroxide and sulfuric acid solutions at the concentration 1 mol dm$^{-3}$ (POCh, Gliwice, Poland).

### 2.2. Irradiation Systems

All irradiation experiments were carried out using UV lamps and solar light simulator.

UV lamps—standard 16AV, (Cobrabid, Poznan, Poland) equipped with two light sources emitting radiation at 254 and 365 nm was used. All examined samples were irradiated by radiation at 365 nm as a representation of natural solar radiation UV-A.

Solar light simulator (SUNTEST CPS+, ATLAS, Champaign, IL, USA) emitting radiation in the range of 300–800 nm was used for experiments in simulated natural conditions.

The intensity of light sources was measured using potassium Reinecke's salt actinometer. The intensity (Es) of radiation emitted by UV lamp was found to be 17.39 W m$^{-2}$ while for the solar light simulator it was 19.53 W m$^{-2}$.

### 2.3. Absorbance Measurements

Monitoring of the current concentration of DOX was carried out spectrophotometically by reading the absorbance at 246 nm. For qualitative assessment of changes in DOX concentration, a calibration plot (ABS = $5.2 \times 10^4 \pm 1.4 \times 10^2$ (DOX) + $0.8 \times 10^{-2} \pm 4.1 \times 10^{-3}$, $r^2 = 0.999$, where ABS—absorbance, (DOX)—concentration of DOX in mol dm$^{-3}$) was constructed for concentrations in the range $10^{-6}$–$2.0 \times 10^{-5}$ mol dm$^{-3}$. The developed spectrophotometric method of DOX determination was characterized by low LOQ and LOD values equal $7.7 \times 10^{-7}$ and $2.3 \times 10^{-7}$ mol dm$^{-3}$, respectively. All spectrophotometric measurements were conducted with a Hitachi U-2800A spectrophotometer (Hitachi High-Technologies Europe GmbH (Mannheim Office), Mannheim, Germany). The following working settings of the device were used: scan speed 1200 nm min$^{-1}$ and spectral bandwidth 1.5 nm.

### 2.4. Experimental Procedures

All irradiation experiments were conducted in a crystallization dish with 100 mL capacity with surface area open to atmosphere.

### 2.5. Direct Photolysis

50 milliliters of working solution of DOX at the concentration of $2.0 \times 10^{-5}$ mol dm$^{-3}$ was subjected to irradiation by a UV-lamp emitting radiation at 336 nm or to solar light in a solar simulator chamber. The spectrum of the solution was recorded every 10 min. A mixture of reagents without DOX irradiated at the same period of time was applied as a blank.

The pH of the aqueous solution was adjusted with 0.1 mol dm$^{-3}$ H$_2$SO$_4$ or 0.1 mol dm$^{-3}$ NaOH. pH was measured with an Elmetron CP-501 pH-meter (produced by ELMETRON, Zabrze, Poland) equipped with a pH-electrode EPS-1 (ELMETRON, Zabrze, Poland). The examination of photolysis in the environmental condition was done in the same manner as described above, using samples of surface water as a solvent.

### 2.6. $H_2O_2$—Assisted Photodegradation Process

$H_2O_2$-assisted photodegradation was studied using a working solution of DOX at the concentration $2.0 \times 10^{-5}$ mol dm$^{-3}$. For this purpose, an appropriate volume of DOX aqueous solution was mixed with varying volumes of hydrogen peroxide so as to obtain final concentration of the oxidant in the range $10^{-1}$–$10^{-3}$ mol dm$^{-3}$. The pH of prepared mixtures was adjusted by adding a proper portion of NaOH or $H_2SO_4$ solution at the concentration 0.1 mol dm$^{-3}$. Mixtures prepared in this way were thereafter subjected to irradiation by UV lamp ($\lambda$ = 365 nm) for 120 min. The spectrum of the reaction solution was recorded every 10 min using the irradiated mixture of reagents without DOX as a blank.

### 2.7. Fenton and Photo-Fenton Processes

The run of Fenton of photo-Fenton process was studied using an aqueous solution of DOX at concentration $2.0 \times 10^{-5}$ mol dm$^{-3}$. For this purpose, a volume of 50 mL of working DOX solution acidified to an optimal pH by 0.1 mol dm$^{-3}$ $H_2SO_4$ solution was mixed with variable volumes of $H_2O_2$ ($10^{-2}$ mol dm$^{-3}$) and $FeSO_4$ ($10^{-2}$ mol dm$^{-3}$). The molar ratio of Fenton reagent ingredients was kept 1:1, and their final concentrations were $10^{-4}$, $5 \times 10^{-5}$, $10^{-5}$ and $5 \times 10^{-6}$ mol dm$^{-3}$. Every 10 min, the spectrum of the reaction mixture was recorded against the mixture of reagents without DOX as a blank.

In the case of examination of the photo-Fenton process, the prepared mixtures were subjected to irradiation by UV light at 365 nm.

### 2.8. Photo Sulfite System

The following procedure was applied: initially, 0.456 mL of the doxazosin standard solution at the concentration of $2.0 \times 10^{-3}$ mol dm$^{-3}$ was introduced into a 50 mL volumetric flask. Next, a small volume of water was added followed by the introduction of 1 mL of ferric sulphate (VI) at the concentration of $2.5 \times 10^{-3}$ mol dm$^{-3}$ and 1 mL of sodium sulphite at the concentration of 0.05 mol dm$^{-3}$. After adding individual reagents, the 50 mL flask was filled to the mark with Milli-Q water. The prepared mixture was then subject to the irradiation by simulated solar light or UV light at $\lambda$ = 365 nm. Like previously, the spectrum of the irradiated mixture was recorded using the irradiated mixture of reagents without DOX as a blank.

## 3. Results and Discussion

### 3.1. Initial Studies

At the beginning of the performed experiments, a UV spectrum of doxazosin aqueous solution was recorded. Its spectral characteristics possessed three distinct maxima: sharp and intense at 196 and 246 nm, and broad and less intense at 328 nm (Figure 2). The kinetics of doxazosin decay was observed by monitoring the changes at 246 nm.

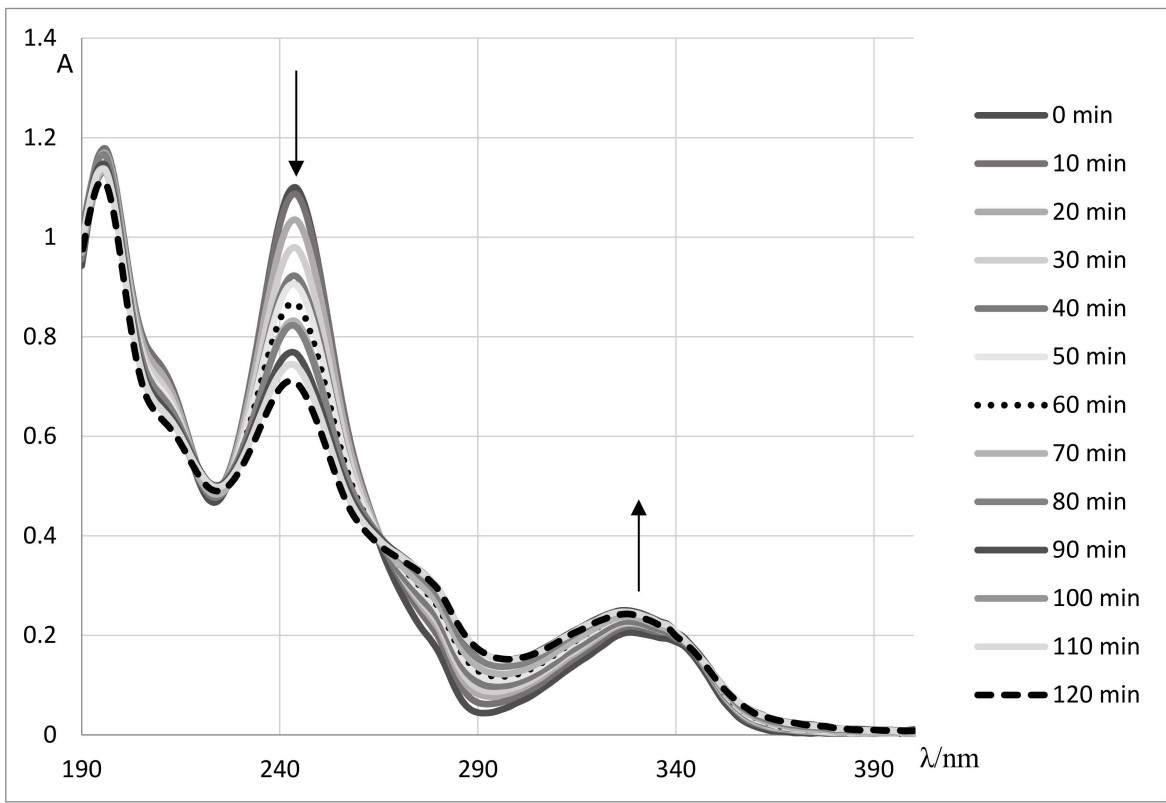

**Figure 2.** The changes in UV spectrum of aqueous doxazosin solution (c = $2.0 \times 10^{-5}$ mol dm$^{-3}$) subjected to irradiation by simulated solar light at native pH 5.56 versus MilliQ water as a blank.

### 3.2. Direct Photolysis in Laboratory Conditions

The photostability of doxazosin in laboratory conditions was checked first. For this purpose, a portion of 50 mL of an aqueous solution of DOX at the concentration $2.0 \times 10^{-5}$ mol dm$^{-3}$ was subjected to irradiation by UV ($\lambda$ = 365 nm) or simulated solar light. It was stated that DOX is a photoliable compound. The following changes in its spectral characteristics were observed: the intensity of the band at 246 nm was gradually decreased while at 328 nm was growing (Figure 2). The rate of the DOX photodegradation process is affected by the type of light, pH of the reaction solution, and the kind of accompanied matrix. It was observed that the process of direct photolysis is more evident under the influence of UV radiation. The obtained results are gathered in Table 1. Additionally, it was noticed that basic pH promotes DOX decomposition, presumably due to the hydrolysis process. Stability experiments were performed in order to confirm this assumption. For this purpose, a series of aqueous solutions of DOX with different pH (in range 1–13) were prepared and thermostated in 0, 30 and 80 °C for 6 h. The used test tubes were wrapped with aluminum foil in order to protect them against light. The UV spectrum of an examined solution was recorded every 20 minutes. It was stated that DOX is stable in an absence of stressed conditions in acidic, neutral and basic medium [20]. Slight signs of hydrolysis were observed in basic solutions heated at the temperature of 80 °C. The assayed value of the hydrolysis rate constant at a temperature of 80 °C and in a basic medium was equal to $3 \times 10^{-5}$ min$^{-1}$.

**Table 1.** The kinetics parameters of doxazosin photodegradation in laboratory solutions and in the presence of natural matrix.

| Studied Process | Used Irradiation | | pH | $k/\text{min}^{-1}$ | $t_{1/2}/\text{min}$ | % of Degradation |
|---|---|---|---|---|---|---|
| **Direct photolysis** | UV 365 nm | | 9 | $2.2 \times 10^{-3}$ | 314 | 24 |
| | UV 254 nm | | 5.56 | $7 \times 10^{-4}$ | 986 | 13 |
| | | | 9.0 | $4.2 \times 10^{-3}$ | 164 | 53 |
| | Suntest | | 5.56 | $4.3 \times 10^{-3}$ (0–60 min) $3.0 \times 10^{-3}$ (61–120 min) | 160 (0–60 min) 230 (61–120 min) | 46 |
| | | | 9.0 | $1.6 \times 10^{-2}$ (0–60 min) $1.7 \times 10^{-3}$ (61–120 min) | 44 (0–60 min) 406 (61–120 min) | 71 |
| **Direct photolysis in presence of natural matrix** | River I | $UV_{365}$ nm | 7.94 | $2.60 \times 10^{-3}$ | 266 | 25 |
| | | Suntest | | $7.0 \times 10^{-3}$ | 98 | 54 |
| | River II | $UV_{365}$ nm | 8.23 | $2.80 \times 10^{-3}$ | 248 | 27 |
| | | Suntest | | $8.5 \times 10^{-3}$ | 82 | 61 |
| | River III | $UV_{365}$ nm | 7.54 | $2.4 \times 10^{-3}$ | 290 | 23 |
| | | Suntest | | $14.5 \times 10^{-3}$ | 48 | 80 |
| | River IV | $UV_{365}$ nm | 7.29 | $3.1 \times 10^{-3}$ | 223 | 29 |
| | | Suntest | | $11.0 \times 10^{-3}$ | 64 | 70 |
| | Carbonate buffer | Suntest | 8.3 | $8.0 \times 10^{-3}$ | 87 | 59 |

### 3.3. Factors Influencing Photolysis of Doxazosin in Environmental Conditions

The stability of doxazosin under simulated environmental conditions was checked next. The goal of this experiment was to answer what the persistence of this compound in natural conditions was. As the chemical composition of natural surface waters is very complex and difficult for reconstruction, real samples of water taken from local rivers were used as solvents for preparing working solutions of DOX. The chemical assessment of the quality of the applied waters showed that the rivers from which the samples were supplied were unpolluted (Table 2). Only in the case of river 3 did the levels of $SO_4^{2-}$ and $NO_3^-$ ions exceed the acceptable reference values. This river flows through agricultural areas and the elevated levels of these ions may be caused by run-offs of fertilizers from fields.

**Table 2.** Chemical characteristics of the used waters.

| Parameter | River 1 | River 2 | River 3 | River 4 | Reference Value | Ref. |
|---|---|---|---|---|---|---|
| | 53°7′ N; 23°7′ E | 53°29′ N; 22°44′ E | 52°20′ N; 23°03′ E | 52°57′ N; 22°57′ E | | |
| pH | 7.94 | 8.23 | 7.54 | 7.29 | 3–11 | 44, 45 |
| Conductivity/μS/cm | 530 | 560 | 330 | 460 | 10–4000 | 46 |
| $SO_4^{2-}$/mg L$^{-1}$ | 15.16 | 77.33 | 116.40 | 14.10 | 10–80 | 47 |
| $NO_3^-$/mg L$^{-1}$ | 70.00 | 22.84 | 21.88 | 35.58 | <50 | 44, 45, 48 |
| $Cl^-$/mg L$^{-1}$ | 41.40 | 10.70 | 199.00 | 35.50 | 0.4–170 | 49 |
| $HCO_3^-$/mval L$^{-1}$ | 5.80 | 5.00 | 4.80 | 5.60 | <14 | 45 |
| Ca/mg L$^{-1}$ | 101.70 | 9.29 | 9.29 | 75.80 | <250 | 50 |
| Mg/mg L$^{-1}$ | 5.98 | 2.82 | 2.57 | 6.20 | <150 | 50 |
| $Fe_{diss}$/mg L$^{-1}$ | 0.33 | 0.23 | 0.04 | 0.77 | <2 | 51 |
| TOC (total organic carbon)/mg L$^{-1}$ | 4.40 | 1.74 | 1.69 | 1.62 | <40 | 52 |
| $O_{2(diss)}$/mg L$^{-1}$ | 10.88 | 54.70 | 37.30 | 15.40 | >4 | 53 |

The photochemical experiments proved that natural waters created an effective chemical system [21]. The observed decomposition rates were similar and strongly dependent on the kind of irradiation used. It was observed that rates of the degradation of DOX under influence of solar light ran two to five times faster than those of the UV-induced process (Table 1) and varied in the range of $7.0 \times 10^{-3}$–$14.5 \times 10^{-3}$ min$^{-1}$. The photolysis experiments with laboratory solutions of DOX in the presence of carbonate ions (pH 8.3) implied that DOX is photoliable compounds and its photodecomposition proceeds mainly via direct photolysis. The created intermediate products, radicals, were initiating a chain process which is inhibited in the presence of radical quenchers such as organic matter or carbonate ions. It was observed that the presence of carbonate and bicarbonate ions did not affect the rate of studied process. This observation allowed us to conclude that the degradation of DOX was independent of changes in the concentration of free radicals in the irradiated solution, but a situation in natural environment was more complicated. This medium is rich in organic matter and a variety of inorganic ions. It is known that dissolved organic matter is photoliable and its products launch a series of reactions with accompanied chemical species [22,23]. If the decomposition of DOX occurs as a result of direct photolysis alone, the rate of its decomposition in the presence of such complex matrix should decrease due to competition for light access. The observed rate of DOX degradation was at least twice higher than this for laboratory solutions. Not so high acceleration of the degradation rate in the presence of the matrix from river 1 could be attributed to a decrease in an energy flux attained by the doxazosin molecules. The acceleration of the decomposition rate, especially visible in the water from river 3, was probably caused by the presence of a variety of inorganic ions which are photosensitive. The photochemical reactions of chloride, nitrate and sulfate ions created a

complicated chain of radical reactions which led to the generation of hydrogen peroxide and hydroxyl, sulfate and nitrate radicals, as well as other radicals [24–27]. The high rate of DOX disappearance in the presence of matrix from the river 3 can be explained as a synergistic action of the system rich in reactive species derived from inorganic ions and organic matter. Chloride radicals can be generated as a result of the direct photolysis:

$$Cl^- + h\nu \rightarrow Cl^{\cdot} + e^- \tag{1}$$

$$Cl^{\cdot} + Cl^- \leftrightarrow Cl_2 \tag{2}$$

or as the result of their interaction with other oxidants, among other excited triplet states of organic sensitizers ($^3$SENS*) [25]:

$$OH^{\cdot} + Cl^- \leftrightarrow HOCl^- \leftrightarrow OH^- + Cl \tag{3}$$

$$HOCl^{\cdot-} \leftrightarrow H^+ \leftrightarrow H_2O + Cl \tag{4}$$

$$HOCl^{\cdot-} + Cl^- \leftrightarrow O^- + Cl_2 \tag{5}$$

$$SO_4^{\cdot} \rightarrow SO_4^{2-} + Cl \tag{6}$$

$$^3SENS^* + Cl^- \rightarrow SENS^{\cdot-} + Cl^{\cdot} \text{ or } ^3SENS^* + 2Cl^- \rightarrow SENS^{\cdot-} + Cl_2 \tag{7}$$

The photochemical reactions of $NO_3^-$ contribute to an increase of the overall concentration of reactive species in the reaction environment as a consequence of the following processes [27,28]:

$$NO_3^- + h\nu \rightarrow NO_3^{-*} \rightarrow NO_2^- + O(^3P) \tag{8}$$

$$\text{or } NO_3^- + h\nu \rightarrow NO_3^{-*} \rightarrow NO_2^{\cdot} + O^{\cdot-} \tag{9}$$

$$O^{\cdot-} + H^+ \leftrightarrow {}^{\cdot}OH \tag{10}$$

The following equilibria are established in the presence of sulfate ions [29]:

$$H^+ + SO_4^{2-} \leftrightarrow HSO_4^- \tag{11}$$

$$SO_4^{2-} + Cl\cdot \leftrightarrow SO_4^{\cdot-} + Cl^- \tag{12}$$

The $HSO_4^-$ ion reacts with OH· radical producing less reactive sulfate radical which however, is involved in the production of more reactive species [29]:

$$SO_4^{\cdot-} + H_2O \rightarrow H^+ + SO_4^{2-} + HO \tag{13}$$

$$SO_4^{\cdot-} + OH^- \rightarrow SO_4^{2-} + HO \tag{14}$$

The acceleration of the decomposition rate of the DOX solution with matrix from river 4 can be assigned to the photo-Fenton process occurring in the presence of dissolved organic matter which is responsible for production of the reactive radicals (HO, O($^3$P), O$^-$, $H_2O_2$) [30,31].

The obtained results pointed out that the decomposition of doxazosin in a natural environment is a very complex process that depends on the chemical composition of an accompanied matrix. It can be stated that the photoreactions of the matrix lead to the increase of the overall concentration of highly reactive radicals such as HO·, which is predominant and mainly responsible for the acceleration of DOX decomposition rate [32].

### 3.4. Kinetics of DOX Decomposition Under Influence of Some Advanced Oxidation Processes

Four advanced oxidation systems: UV/$H_2O_2$, classical, and photocatalytic Fenton reaction, and photo-sulfite systems were chosen from among a number of possible AOPs methods, and their efficiency in DOX degradation were examined. The kinetics of DOX decomposition under the influence of UV/$H_2O_2$, system was studied first. The influence of an oxidant concentration in the range $10^{-4}$–$10^{-2}$

mol dm$^{-3}$ and pH (3.85–8) was checked. It was stated that the degradation of DOX in the UV/H$_2$O$_2$ system fit the pseudo-first order reaction. An addition of hydrogen peroxide caused a three-to-eight-fold increase in the reaction rate in comparison to the direct photolysis process. The observed enhancement depends on the used light, concentration of oxidant, and pH (Tables 1 and 3). It was noted that the increase in hydrogen peroxide concentration increased the reaction rate, but this augmentation was rather slight. The 100-fold reinforcement in the oxidant concentration amplified the reaction rate by only 7%. Analogically, as in the case of direct photolysis, the basic pH promoted the studied process, as the acidic pH of the rate of reaction was almost negligible. The role of the oxidant and light was checked next. For this purpose, two series of DOX aqueous solutions at pH 8 with the concentrations $2.0 \times 10^{-5}$ mol dm$^{-3}$ and $2.5 \times 10^{-5}$ mol dm$^{-3}$ were prepared. Appropriate volumes of H$_2$O$_2$ working solutions were added to each test tube so that the concentration of the oxidant was in the range $10^{-5}$–$5 \times 10^{-4}$ mol dm$^{-3}$. Each test tube was wrapped with aluminum foil in order to protect against light and subsequently left at ambient temperature for 24 h. Thereafter, the spectra of each mixture were recorded. No changes in the spectral characteristics of doxazosin were observed, which proves the important dual role of light in this process. On the one hand, light induces the process of direct DOX photolysis, while on the other hand it breaks down the dihydrogen peroxide into hydroxyl radicals according to the reaction [33]:

$$H_2O_2 + h\nu \rightarrow 2OH\cdot \tag{15}$$

Considering the above findings, it could be concluded that the observed enhancement in the rate of DOX degradation is a sum of the two above processes.

The classical Fenton system consisting of a solution of inorganic ferrous salt and hydrogen peroxide was examined next. The operation of the Fenton system is very complex and not fully known yet. The reaction mechanism involves the generation of hydroxyl radical according to the following reaction) [34]:

$$Fe^{2+} + H_2O_2 \rightarrow Fe^{3+} + OH\cdot + OH^- \tag{16}$$

Its efficiency in the degradation of organic pollutants is affected by the concentration of reagents, their molar ratio, and pH of reaction medium. In order to select the optimal concentrations of Fenton reagent components allowed to follow DOX degradation kinetics, a series of experiments were carried out using different concentrations of ingredients at their molar ratio of 1: 1. For this purpose, the concentrations in the range $5 \times 10^{-6}$–$5 \times 10^{-4}$ mol dm$^{-3}$ were applied. The second order kinetics was assumed for studied Fenton and photo-Fenton systems. It was stated that at the lowest examined concentration, the observed process proceeded too slowly, but when the highest one was used, the total disappearance of the drug was observed in five minutes after the initiation of the reaction. The Fenton reagent with the concentration of $10^{-4}$ mol dm$^{-3}$ of the components was selected for further testing, resulting in the determination of the kinetic parameters of DOX degradation with good precision. The influence of pH was checked next; it is known that the optimal working pH for the studied process is contained in the range 2–5 [34]. At an excess of hydrogen ions, less reactive positively charged ferrous species are formed [35]. Additionally, the surplus of hydrogen ions act as radical scavengers according to the following reaction:

$$H^+ + \cdot OH \rightarrow H_2O \tag{17}$$

The precipitation of Fe(II) and Fe(III) hydroxides is observed at pH > 5. In order to select the best pH for the DOX degradation, rates of reaction at various pH values in the range 2–5 were measured. The obtained output showed that pH 3.5 was optimal for the studied process.

The influence of molar ratio of the Fenton reagent constituents on kinetics of DOX degradation was checked. The following molar ratios of $n_{H2O2}$:$n_{Fe(III)}$ = 1:1, 2:1, 10:1, and 1:2, 1:5, 1:10 were used. The obtained results demonstrated (Table 3) that the use of an excess of hydrogen peroxide in ratio to ferrous ions promotes the degradation process while the reverse ratio is unfavorable for the course

of the reaction. Following this, the effect of UV radiation on the course of the DOX decomposition reaction with the Fenton reagent was examined. The results shown in Table 3 proved that application of light enhanced the efficiency of the Fenton process due to the following process [34]:

$$Fe^{3+} + h\nu + H_2O \rightarrow Fe^{2+} + \cdot OH + H^+ \tag{18}$$

The regeneration of ferrous ions increased the overall concentration of hydroxide radicals and slowed down an increase in pH of reaction medium [34].

Recently, the use of sulphate radicals to remove organic compounds from waters has attracted more attention due to their stability and high oxidation potential (2.5–3.1 V vs. NHE) [36]. Additionally, they can work in a wide range of pH, so a rigid control of this parameter is not necessary [37,38]. The prevailing type of oxidant in the reaction environment depends on value of the initial pH. It was found that $SO_4\cdot^-$ radicals predominate at the acidic pH, while at the basic pH $\cdot OH$ radicals are responsible for oxidation of organic compounds [37,38]. The only disadvantage of using sulphate radicals is the necessity to activate precursors to obtain the right concentration of the oxidant [37,38]. An alternative to conventional methods for production of $SO_4\cdot^-$ radicals ferric sulphate-sodium sulphite system in the presence of light was proposed [39]. This system is based on Fe-catalyzed sulphite oxidation and photochemical cycle of Fe(III)-Fe(II) species. For this reason, it can be considered a modification of the Fenton system [39]:

$$Fe^{3+} + HSO_3^- \leftrightarrow FeSO_3^+ + H^+ \tag{19}$$

$$FeSO_3^+ \rightarrow Fe^{2+} + SO_3\cdot^- \tag{20}$$

$$SO_3\cdot^- + O_2 \rightarrow SO_5\cdot^- \tag{21}$$

$$SO_5\cdot^- + HSO_3^- \rightarrow SO_4\cdot^- + SO_4{}^{2-} \tag{22}$$

$$SO_5\cdot^- + SO_5\cdot^- \leftrightarrow 2SO_4\cdot^- + O_2 \tag{23}$$

$$SO_5\cdot^- + SO_5\cdot^- \leftrightarrow SO_3\cdot^- + HSO_5^- \tag{24}$$

$$Fe^{2+} + HSO_5^- \rightarrow SO_4\cdot^- + Fe^{3+} + OH^- \tag{25}$$

$$FeSO_3^+ + light \rightarrow Fe^{2+} + SO_3\cdot^- \tag{26}$$

$$FeOH^{2+} + light \rightarrow Fe^{2+} + \cdot OH \tag{27}$$

The results of the above chain of reactions is a mixture of a variety of radicals where $SO_4\cdot^-$ and $\cdot OH$ are predominant [39].

The kinetics of DOX degradation under the influence of UV/Vis-Fe(III)-sulphite system was examined. For this purpose, a series of DOX solutions at concentration $2.0 \times 10^{-5}$ mol dm$^{-3}$ were mixed with variable volumes of ferric sulphate solution at the concentration $2.5 \times 10^{-3}$ mol dm$^{-3}$ and sodium sulphite at the concentration $5 \times 10^{-2}$ mol dm$^{-3}$. The applied concentrations of reagents are shown in Table 3.

The obtained results showed that the efficiency of light- Fe(III)-sulphite system depends on the molar ratio of reagents and the applied light. It was observed that the use of 10-fold excess of $Na_2SO_3$ in ratio to $Fe_2(SO_4)_3$ and irradiation by solar light resulted in total DOX decomposition in 90 minutes. In order to recognize the main oxidizing agent in the light- Fe(III)-sulphite system the scavenging experiments for the degradation of DOX were performed by adding tert-butyl alcohol (TBA) to the reaction medium. The applied final concentration of TBA was 0.5 mol dm$^{-3}$ while ferric sulphate and sodium sulphite were $10^{-3}$, respectively. The kinetic graphs of DOX concentration changes without the presence of TBA outlined in Figure 3, show that DOX degradation by the Fe(III)-sulphite process is mainly caused by sulphate radicals' oxidative action. At the presence of tert-butyl alcohol, the lesser extent of DOX decay was achieved by approximately 20%. This effect was particularly pronounced in an initial stage of the reaction. As the rate of TBA reaction with hydroxyl radicals is approximately

1000-fold greater than that with sulphate radicals [39], it could be concluded that $SO_4^{\cdot-}$ radicals are the major reactive species responsible for DOX degradation.

**Table 3.** Kinetic parameters of DOX degradation in advanced oxidation systems.

| Studied Process | Concentration of $H_2O_2$/mol dm$^{-3}$ | Concentration of $Fe^{2+}$/mol dm$^{-3}$ | pH | $k$/min$^{-1}$ | $t_{1/2}$/min | % of Degradation |
|---|---|---|---|---|---|---|
| UV/$H_2O_2$ | $5 \times 10^{-4}$ | | | $11.6 \times 10^{-3}$ | 59.7 | 72 |
| | $10^{-4}$ | | | $11.9 \times 10^{-3}$ | 58.5 | 73 |
| | $10^{-2}$ | - | 8 | $12.10 \times 10^{-3}$ | 57.2 | 73.5 |
| | $5 \times 10^{-2}$ | | | $12.50 \times 10^{-3}$ | 55.5 | 74.5 |
| | | | | $k$/min$^{-1}$mol$^{-1}$ dm$^3$ | | |
| Classical Fenton reaction | $10^{-4}$ | $10^{-4}$ | | 52.5 | 982 | 15 |
| | $2 \times 10^{-4}$ | $10^{-4}$ | | 127.6 | 535 | 27 |
| | $10 \times 10^{-4}$ | $10^{-4}$ | | 332.8 | 200 | 48 |
| | $10^{-4}$ | $2 \times 10^{-4}$ | | 51.2 | 956 | 12 |
| | $10^{-4}$ | $5 \times 10^{-4}$ | | 45.0 | 1351 | 11 |
| | $10^{-4}$ | $10 \times 10^{-4}$ | | 5.5 | 9823 | 1.5 |
| Photo-Fenton reaction | $10^{-4}$ | $10^{-4}$ | | 86.6 | 657 | 25 |
| | $2 \times 10^{-4}$ | $10^{-4}$ | | 296.0 | 244 | 46 |
| | $10 \times 10^{-4}$ | $10^{-4}$ | | 3308.0 | 31 | 100 |
| | $10^{-4}$ | $2 \times 10^{-4}$ | 3.5 | 71.8 | 785 | 21 |
| | $10^{-4}$ | $5 \times 10^{-4}$ | | 265.0 | 239 | 50 |
| | $10^{-4}$ | $10 \times 10^{-4}$ | | 53.0 | 1002 | 17 |
| | Concentration of $Fe_2(SO_4)_3$/mol dm$^{-3}$ | Concentration of $Na_2SO_3$/mol dm$^{-3}$ | | | | |
| UV/Fe(III)-$SO_3^{2-}$ | $5 \times 10^{-5}$ | $10^{-3}$ | | 2538 | 22 | 61 |
| | $5 \times 10^{-5}$ | $2 \times 10^{-3}$ | | 1324 | 41 | 63 |
| | $5 \times 10^{-5}$ | $3 \times 10^{-3}$ | | 715 | 77 | 59 |
| | $5 \times 10^{-5}$ | $4 \times 10^{-3}$ | | 394 | 58 | 58 |
| | $10^{-4}$ | $10^{-3}$ | | 7892 | 7 | 75 |
| | $1.5 \times 10^{-4}$ | $10^{-3}$ | | 587 | 93 | 66 |
| | $2 \times 10^{-4}$ | $10^{-3}$ | | 544 | 100 | 64 |
| Vis/Fe(III)-$SO_3^2$ | $10^{-4}$ | $10^{-3}$ | | 1986 | 3 | 100 |
| Vis/Fe(III)-$SO_3^2$/TBA | $10^{-4}$ | $10^{-3}$ | | | | 75 |

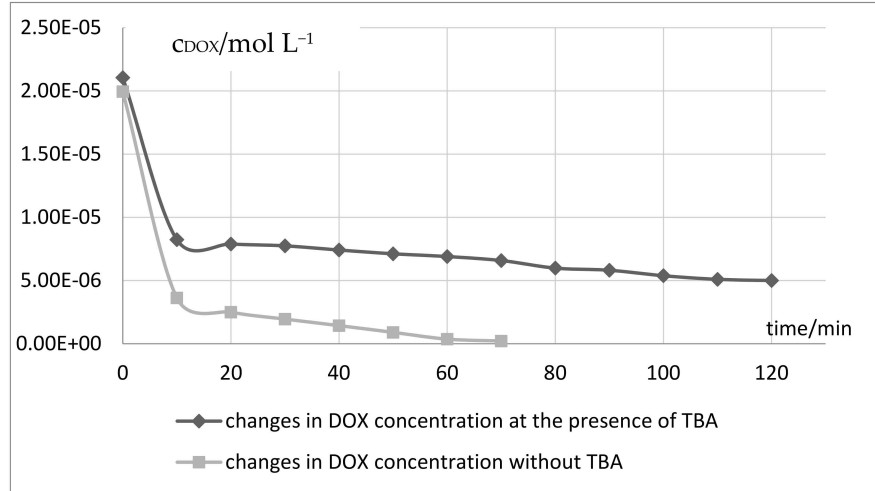

**Figure 3.** Changes in DOX concentration under influence of Vis-Fe(III)-sulphite system in presence and without TBA.

### 3.5. A DFT Mechanistic Study of the DOX Decomposition

In order to gain a deeper understanding of the initial stages of the process of molecular degradation of doxazosin, a quantitative DFT mechanistic study was carried out. The calculations were performed with the GAUSSIAN 09 program [40]. To accurately capture properties of the transition states, a DFT functional M06-2X [41], designed especially for the chemical kinetics, was utilized in combination with a Dunning's correlation consistent basis set cc-pVDZ. This theory level is sufficient to capture the physical change along the reaction coordinate for processes investigated here. To model the solvent (water) effect, the CPCM approach was used [42], as implemented in the G09 program. The reaction course is outlined in Figure 4, and all the species involved are pictured in Figure 5. As seen from Figure 4, the reaction is initialized by the addition of the –OH group to the aromatic carbon (atom 7). It is well known that such processes go through short-lived intermediate (thermodynamically controlled step) and transition states (kinetic control) to form an adduct vulnerable to further degradation. It starts with breaking the bond between the addition center and neighboring nitrogen (N10), which is followed by the internal H transfer to form an intermediate (IM3) with keto and amino groups on the bond breaking sites (atoms C7 and N10). The actual destruction of the molecular structure occurs via breaking bond between N3 and C7. Recently, a scheme of the DOX decomposition based on the B3LYP results was proposed [43]. The favorable pathway takes place by cracking bonds N12-C15 and N12-C16. We tried to reproduce this scheme; however, at the M06-2X/cc-pVDZ theory level, no proper transition states leading to ring degradation were found despite many attempts. Nevertheless, since the molecule break-up proposed here starts at the closest vicinity of the bonds N12-C15 and N12-C16 (see Figure 5), our results led to the products with similar molecular masses as those proposed in [43]. As such, their experimental analysis supports both proposed pathways.

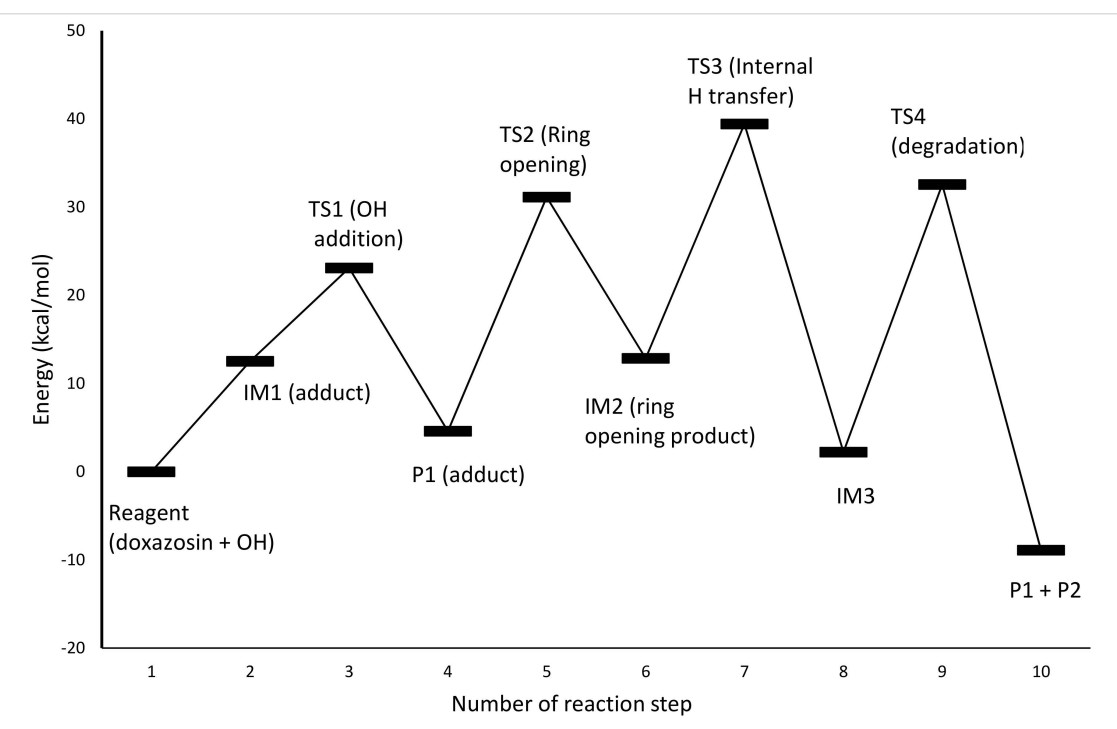

**Figure 4.** Energy profile of the proposed reaction mechanism (in terms of ΔG values, kcal/mol).

**Figure 5.** The optimized (at the M06-2X/cc-pVDZ level) geometries of the species involved in the reaction scheme from Figure 4.

## 4. Conclusions

The obtained results show that doxazosin is a photoliable compound. Experiments done with laboratory solutions demonstrated that DOX direct photolysis is promoted by the basic medium and proceeds faster under the influence of UV light. It was stated that observed degradation of DOX is the

result of direct photolysis. The presence of natural matrix acted as a photosensitizer and accelerated the degradation process. The presented surface waters natural radicals made DOX more sensitive to visible light. The run of DOX degradation under influence of some AOP-s were examined. Among the processes taken under consideration, photo-Fenton reaction and Vis/Fe(III)-SO$_3$$^{2-}$ appeared to be the most efficient.

The DFT mechanistic study provided an understanding of the role of OH$^-$ ion in the photolysis process and pointed out which transformations of molecule lead to its decomposition. It was stated that the initial step of the process is the formation of unstable adduct by bonding the −OH group to the C7 carbon of the aromatic ring. The calculation pointed to the dissociation of the bond between N3 and C7, which lead to molecule disintegration, followed by the internal H transfer and formation of the IM3 intermediate.

**Author Contributions:** Conceptualization, J.K. (Joanna Karpinska) and A.S.; methodology J.K. (Joanna Karpinska) and A.S.; DFT study, A.R.; investigation, J.K. (Jolanta Koldys), and A.S.; writing—J.K. (Joanna Karpinska), A.R.; writing—review and editing, J.K. (Joanna Karpinska) and A.S.; visualization, A.R. and A.S.; supervision, J.K. (Joanna Karpinska).

**Funding:** This research received no external funding.

**Acknowledgments:** The authors would like to thank the Computational Center of the University of Bialystok (Grant GO-008) for providing access to the supercomputer resources and the GAUSSIAN 09 program.

**Conflicts of Interest:** The authors declare no conflict of interest.

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
