# Peer review of "Studies on the Kinetics of Doxazosin Degradation in Simulated Environmental Conditions and Selected Advanced Oxidation Processes"

_water, doi:10.3390/w11051001_

Round 1
Reviewer 1 Report
The manuscript describes a study of photo-degradation of doxazosin in aqueous solutions. The degradation rates were determined at the presence of different oxidizing agents and irradiation of UV light of specific wavelengths or simulated solar light. The use of water from natural resources as media of the degradation reactions has values in better mimicking the natural degradation of the compound.
The written English language has to be improved. There are many inappropriate uses of words and grammar issues, which make the text difficult to read. The introduction is lengthy and so are some of the conceptual descriptions later in the text.
In addition, I have several comments/questions regarding the scientific content of the manuscript as follows.
(1) What are the power (in watts) of UV (365 or 254 nm) and the power of solar light used in the study? The statement in line 187 and 188 is true only when the two types of light were applied at the same power. But this information was not given in the text.
(2) Line 110, “… irradiation by UV-lamp emitting radiation at 336 nm…,” should it be 365nm?
(3) The placement of Table 3 should be before Figure 3 since Fig. 3 is referred first in the text.
(4) There was no definition of visible light in the text. Is it the same as the simulated solar light described in the materials and methods? Then the change of terms makes it a bit confusing in the last two pages of the manuscript.
(5) Figure 4, there is no x-axis or coordination.
(6) In Figure 5., line-angle structures should be given for each molecular ball-stick model. It is easier for readers to understand the reaction mechanism investigated in this work.
Author Response
Summary of changes in the manuscript
Reviewer: 1
Comment (1): The values of intensity of radiation emitted by UV lamp and the solar light were given.
Comment (2): The value of wave of emitted light was corrected it is equal 365 nm.
Comment (3): The placement of Table 3 was changed. Now it is before Figure 3.
Comment (4): In order to maintain the same reproducible conditions, in all experiments with Vis light were used the light emitted by the solar-light simulator
Comment(5): Figure 4, there is no x-axis or coordination.
Reply: The values on the x-axis indicates a number of reaction step. According to the suggestion, Figure 4 was updated accordingly.
Comment(2): In Figure 5., line-angle structures should be given for each molecular ball-stick model. It is easier for readers to understand the reaction mechanism investigated in this work.
Reply: Indeed, line angle structures are far more readable than ball-stick models. Consequently, Figure 5 was redrawn as suggested.
Reviewer 2 Report
Page 2
Line 63: has not been repoted
Page 3
Line 85: Other reagents used were:
Page 5
Line 176: this experiment was to answer the question
Page 6
Line 177 : As the
Line 8181: 3 that the levels of SO4 and NO3 ions exceeded
Line 186: strongly dependent
Line 195: of the changes
Page 8
Line 267 known
Line 272:to follow DOX degradation kinetics
Line 279: testing, it resulted in the determination of the kinetic parameters
Line 282: for the studeied process
Line 292: Following this, the effect of UV radiation
Line 293: The results shown in Table 3
Line 299: has attracted more
Line 313 Equation 21 is NOT balanced
Line 351: Equation 23 is NOT balanaced
Page 9
Line 324: are shown in Table 3
Page 10
Line 343: In order to gain a deeper understanding
Line 349: outlined in figure 4
Line 360: despite many attempts
Line: 375: The natural radicals present in surface waters made DOX more sensitive
Line 376/377: Among the processes considered
Author Response
Summary of changes in the manuscript
All grammar and spelling mistakes indicated by the reviewer have been corrected . Done as suggested.
Round 2
Reviewer 1 Report
(1)Professional English editing is needed.
(2) Some structures in Figure 5 are not drawn correctly, especially those carbons bonded to nitrogen through a triple bond; they do not meet the octet configuration.
Author Response
Prof. Amelia Yan, Water
Professor Amelia Yan, Water
Manuscript ID: water-486953
Dear Prof. Yan,
Thank you for forwarding the reviewers’ comments. We would like to thank the reviewer for her/his careful reading of the manuscript and helpful comments. We have revised the manuscript accordingly. Below is the summary of our changes.
We hope that these changes are sufficient and would like to thank you for your consideration of publishing the manuscript in Water.
Sincerely yours,
Joanna Karpińska
Summary of changes in the manuscript
Reviewer: 1
Comment (1)Professional English editing is needed.:
Reply: The manuscript has been revised by native speaker and corrected.
Comment (2): Some structures in Figure 5 are not drawn correctly, especially those carbons bonded to nitrogen through a triple bond; they do not meet the octet configuration.
Reply: All the structures from Figure 5, except for reactants (doxazosin + OH), are highly reactive intermediates and/or transition states, for which the octet rule may not be satisfied for all atoms. In the case pointed by Reviewer 1, the triple bonds were incorrectly assigned by the molecular editor (Chemcraft). These errors and some others were manually fixed.